# Determination of Optimum Processing Condition of High Protein Laver Chip Using Air-Frying and Reaction Flavor Technologies

**DOI:** 10.3390/foods12244450

**Published:** 2023-12-12

**Authors:** Gyeong-Tae Jeong, Changheon Lee, Eunsong Cha, Seungmin Moon, Yong-Jun Cha, Daeung Yu

**Affiliations:** 1Interdisciplinary Program in Senior Human Ecology, Changwon National University, Changwon 51140, Republic of Korea; rudxo0118l@naver.com (G.-T.J.); ckdgjs3306@changwon.ac.kr (C.L.); ssfd2420@naver.com (E.C.); 980716_@naver.com (S.M.); 2Department of Food and Nutrition, Changwon National University, Changwon 51140, Republic of Korea; yjcha@changwon.ac.kr

**Keywords:** laver, air-flying, reaction flavor, surimi, protein

## Abstract

This study aimed to develop a high-protein and gluten-free laver chip using air-frying and reaction flavor technologies via response surface methodology (RSM). The optimum processing condition (*w*/*w*) was determined with a batter composition of 20% dried laver, 21.3% hair tail surimi, and 58.7% rice flour. Additional ingredients included б-gluconolactone, NaHCO_3_, soybean oil, corn syrup, table salt, saccharin, and a mixture of distilled water and reaction flavor-inducing solution (RFIS). The laver pellet processed and dried (50 °C, 1–2 h) with air-frying (195 °C, 52.5 s) to process the laver chip. The values of brittleness and puffing ratio of the laver chip were 6.93 ± 0.33 N and 116.19 ± 0.48%, respectively, with an error within 10% of the predicted values of RSM. RFIS was prepared via RSM with the addition of precursor substances (*w*/*v*) of methionine 0.54%, threonine 3.30%, glycine 2.40%, glutamic acid 0.90%, and glucose 3% to distilled water and then heating reaction (121 °C, 90 min). The quantitatively descriptive analysis (QDA) of RFIS, baked potato-like and savory odor were 6.00 ± 0.78 and 4.00 ± 0.91, respectively, with an error within 10% of the predicted values. The laver chip exhibited high-protein (24.26 ± 0.10 g%) and low-calorie (371.56 kcal) contents.

## 1. Introduction

Processed laver is the best export item for South Korea seafood, with exports reaching $ 580 million in 2019 (MOF, 2020). Until recently, the overseas export of laver processed foods was mainly focused on seasoned laver [1]. However, as laver snack products continue to be developed, the trend is diversifying. Among the laver-based snack products, the most representative is *kim bugak* (laver snack), a traditional Korean food [2]. However, these *kim bugak* products are associated with problems such as rancidity, quality degradation, etc., during distribution due to the excessive fat caused by oil frying [3].

Laver (*Porphyra yezoensis*), a type of red algae, is mainly cultivated in the southwestern coast of South Korea, including Wando, Jeollanam-do, and Jeju Island [4]. Dried laver contains more than 30% protein and less than 1% fat. Among the fat, the relative content of unsaturated fatty acids is more than three-times higher than that of saturated fatty acids, and the content of polyunsaturated fatty acids is more than 60%. Therefore, dried laver is highly valued as a health food [5]. In addition, laver is an alkaline food with relatively high levels of minerals such as calcium, potassium, iron, and iodine. Therefore, it was reported that Koreans who frequently consume laver have a much lower incidence of iodine deficiency than Westerners [6]. In particular, porphyran, a water-soluble acidic polysaccharide found in laver, has been shown to promote bowel movement, reduce the time that harmful substances remain in the intestines, and increase stool output, thereby reducing the incidence of colorectal cancer [5]. In addition, it inhibits the rise of cholesterol and blood sugar, which are the causes of atherosclerosis, and has been shown to have excellent antioxidant activity [7].

Snacks and other processed laver products that have been sold on the market are oil fried at high temperatures during processing. This can lead to the formation of oxidized substances in the frying oil, which can negatively affect the taste and flavor of the product, leading to a decrease in overall quality [8]. In addition, the oxidation of the oil can shorten the shelf life of the product [3]. Moreover, the consumption frequency of oil fried and fast foods with high fat content has increased due to the influence of Westernized dietary habits. This is recognized as one of the factors causing obesity and chronic diseases, and recently, the preference for low-fat products is increasing [3]. In addition, most of these products are made with wheat flour as the main ingredient. As a result, from the perspective of modern dietary habits that pursue diet and health, consumers who prefer gluten-free products are increasing, along with the tendency to reduce the intake of gluten, which is derived from wheat flour and causes allergies [9].

Air-frying technology is a cooking method that uses the principle of heating food by circulating hot air (maximum 200 °C) generated by heating elements inside the appliance with an internal fan [10]. This air-frying technology can produce crispy fried dishes without deep-oil frying, as the moisture evaporates quickly. At the same time, the oil inside the cooked food is also externally released and removed [10]. Products developed using this air-frying technology have been reported including plant-based products, such as dried lotus root chips [11], sweet potato snacks [12], banana chips [13], and falafel made from beans [14], as well as animal-based products, such as fish snacks [15], surimi products [16], and tilapia skin snacks [17]. The common points of these studies are that they have a lower fat content than traditional deep-oil frying methods, are primarily aimed at health, and most of them have identified the optimum processing conditions using response surface methodology.

On the other hand, reaction flavor technology is a type of Maillard reaction in which the sugar–amino reaction of precursor materials, such as amino acids and monosaccharides, is processed by heat treatment. This methodology has long been used to enhance the flavor of processed foods [18]. In particular, reaction flavor application technology using heat treatment has been applied to food processing, such as the generation of meat-like flavor [19], the improvement of the flavor of stalked sea squirt drips [20], the production of crab flavor from snow crab cooker effluent [21], and the development of teriyaki sauce through the production of shrimp flavor [22]. In particular, studies have also reported on the factors of various precursor materials affecting the generation of meat flavor [23]. These reaction flavor studies have been conducted to impart savory flavor to the inherent flavor of foods or to mask off-flavors. Therefore, it can be ensured that applying this reaction flavor technology in this study will contribute greatly to the improvement in flavor of laver chips, as well as masking the off-flavors unique to laver.

Therefore, this study aimed to process laver chips using air-frying and reaction flavor technologies and to identify the optimum processing conditions for a new type of laver chip that is high in protein, gluten-free, and low in calories, which differs from conventional oil-fried laver products.

## 2. Materials and Methods

### 2.1. Materials

About 300 g of the dried laver (*Porphyra yezoensis*) was taken and roasted for 5–10 s on a portable gas stove (Blanc, Suntouch Co., Chungnam, Republic of Korea). Then, it was ground for 9 min (ground for 1 min, 2 min rest, 3 times) using a blender (SHMF-3260S; Hanil Electric Co., Bucheon, Republic of Korea) to a volume of 6720× *g*. The ground laver was filtered through a 30 mesh sieve (Jinsungstar Co., Gyeonggi, Republic of Korea), filled in a zip-lock bag (low-density polyethylene (LDPE), 25 cm × 30 cm, Cleanlab Co., Seoul, Republic of Korea) and stored in a freezer (−20 °C) for the experiment.

Surimi (hairtail surimi, Xiangshan shipu goulog aquatic products Co., Ningbo, China) was provided by Manjun Food (MANJUN FOODS Co., LTD., Seoul, Republic of Korea), and was divided into 500 g portions each. It was then placed in a zip-lock bag (low-density polyethylene (LDPE), 15 cm × 10 cm, Cleanlab Co., Seoul, Republic of Korea) and stored in the freezer (−20 °C) for the experiment.

As minor ingredients, б-gluconolactone (GDL, USA product) and sodium bicarbonate (NaHCO_3_, France product) were purchased from ES Food Ingredients (ESfood Co., Gunpo, Republic of Korea) and Bread Garden (Breadgarden Co., Seongnam, Republic of Korea) as a leavening agent (food additive grade). In addition, soybean oil (pure refined soybean oil, Hapyo, Seoul, Republic of Korea), salt (Reproduced salt, Simplus, Jeonnam, Republic of Korea), and rice flour (Hwami rice flour, Hwami, Incheon, Republic of Korea) were purchased from a retail mart located in Changwon, Gyeongnam Province.

On the other hand, amino acids (proline, methionine, threonine, glycine and glutamic acid) and glucose as the precursor substances, which were used for processing the reaction flavor-inducing solution (RFIS), were provided by Vixxol (Vixxol Co., Ltd., Ansan, Republic of Korea) for free and saccharin and corn syrup as food additives were provided by Sungsim Master Food (Sungsim Master Food., Gyeongnam, Republic of Korea) for free.

### 2.2. Preparation of Reaction Flavor-Inducing Solution (RFIS)

RFIS, which was added to laver chip to impart flavor, was prepared according to the method presented in studies by Cha et al. [24], Cha and Wang [25], and Ahn et al. [21]. To process RFIS, 4 types of amino acids (methionine 0.54 g, threonine 3.30 g, glycine 2.40 g and glutamic acid 0.90 g) and glucose (3.00 g) for food additives, selected according to the optimum composition obtained through preliminary experiments and RSM analysis, were added to 100 mL of distilled water and reacted in an autoclave (LAC-5080SD 80L, Daihan labtech Co., LTD., Namyang-ju, Republic of Korea) at 121 °C for 90 min. This RFIS was then added to a laver chip during processing.

### 2.3. Preparation of Laver Chip

The following provides the amount of ingredients used in the production of the laver chip. First, batter was made by mixing roasted laver powder and surimi in a 1:1 ratio and added rice flour (58.7 g) with blending for 1 min with the blender as shown in Figure 1. Then, leavening agents (3 g of GDL and 1.5 g of NaHCO_3_), soybean oil (4 g), and 14 mL of RFIS were added in a distilled water 126 mL in a certain amount according to the condition. The completely dissolved solution was added to the laver and surimi batter and blended again for 1 min.

To process the laver chip of a uniform size, the prepared laver chip batter was placed in a pre-made acrylic mold (150 mm wide × 250 mm long × 1.5 mm thickness) and evenly spread with a wooden rod (ϕ40 mm × 300 mm). Next, the batter was cut to a size of 25 mm wide and 40 mm long using a pre-made rectangular acrylic plate (25 mm wide × 150 mm long). To adjust the moisture content to a suitable level for the laver chip processing (12–14%), the cut batter was dried in a dry oven (OF-22; Jeio Tech, Daejeon-Si, Republic of Korea) at 50 °C for 1~2 h. The dried laver chip pellet was then fried in an air fryer (MC35A8599LE; Samsung Co, Port Klang, Malaysia) preheated to 195 °C for the optimum frying time. During processing, the moisture content of the laver chip was measured using a moisture analyzer (MB-25; OHAUS Corporation, Seoul, Republic of Korea) after drying and heating stages.

### 2.4. Preliminary Experiment for Determining the Optimum Composition Ratio of Laver Chip Batter

A preliminary experiment was conducted in four stages to determine the optimum composition ratio of the batter for processing the laver chip. The composition ratio of the major ingredients (laver and surimi), the ratio of leavening agents (GDL: NaHCO_3_), and the amount of soybean oil were determined to obtain the optimum texture of the laver chip. The amount of corn syrup was determined to obtain the optimum flavor of the laver chip. The optimum composition ratio of these ingredients was determined through brittleness, puffing ratio, and sensory tests.

### 2.5. Determination of Optimum Processing Condition for Processing Laver Chip by Response Surface Methodology (RSM)

The experimental plan for determining the optimum condition in the laver chip processing process was designed according to the central composite design (CCD) of the RSM using the Minitab Statistical Software program (Version 19, State College, PA, USA), and the optimum condition was determined.

#### 2.5.1. Determination of Optimum Physical Condition for Processing Laver Chip by RSM

In the laver chip manufacturing process, the amount of surimi was set to 1 to 1.8 times (*w*/*w*) of the amount of the laver, which was obtained through preliminary experiments. This is because the texture of the laver chip hardened as the amount of laver added increased, and the consumers would not prefer the dark black color of the laver itself. Therefore, the amount of laver was fixed at 20% (*w*/*w*), excluding moisture. The rice flour content (%) was set to [100 g—(laver powder content + surimi content)] as shown in Table 1 (RSM 1). In addition, the independent variables of the air fryer heating temperature and time were set through preliminary experiments to determine the time range that the laver chip pellet did not burn when processed at 195 °C. Both coded and uncoded actual experimental values are presented in Table 1 according to the experimental range. The experiment was conducted in 14 runs according to CCD, and the dependent variables were set to the puffing ratio and brittleness, which affect the texture of the laver chip.

#### 2.5.2. Determination of Optimum Mixing Ratio of RFIS for Processing Laver Chip by RSM

As shown in Table 1 (RSM 2), among the three additives, which were identified by preliminary experiments including sensory evaluation and studies of Cha [24], Cha and Wang [25], and Ahn et al. [21]. The optimal determination for this experiment was obtained by selecting 3 compounds (methionine, threonine, and glycine) that have a great influence on the flavor as independent variables of RSM. Glutamic acid and glucose were fixed at 0.3% and 1.0% (*w*/*v*), respectively, and the remaining three amino acids (methionine, threonine, and glycine) were designated as independent variables through the RSM for identifying the optimization condition of RFIS. The added amount of amino acids (X_1_: methionine, X_2_: threonine, X_3_: glycine) was set, and they were coded as −2, −1, 0, +1, and +2, respectively. The experimental points were 17 in total, including 8 cube points (±α, 2), 6 axial points (±1), and 3 central points (0).

### 2.6. Proximate Composition of Raw Materials and Laver Chip

The proximate composition of raw materials (laver, surimi, and rice flour) and laver chip was determined according to the AOAC method [26] as follows: moisture content by oven drying, crude protein by semi-micro Kjeldahl method, crude fat by Soxhlet extraction, and crude ash by dry ashing. The results of the proximate composition of the laver were that the moisture content was 1.45 ± 0.04%, carbohydrate was 46.76 ± 0.20, crude protein was 42.83 ± 0.06%, crude fat and crude ash were 0.80 ± 0.09% and 8.16 ± 0.01%, respectively. Regarding surimi, the moisture content was 75.52 ± 0.08%, carbohydrate was 4.92 ± 1.33, crude protein was 17.83 ± 1.15%, crude fat and crude ash were 1.26 ± 0.10% and 0.47 ± 0.00%, respectively. With regard to the rice flour, the moisture content was 7.21 ± 0.15%, carbohydrate was 85.06 ± 0.36, crude protein was 6.27 ± 0.09%, crude fat and crude ash were 0.37 ± 0.06% and 1.09 ± 0.06%, respectively.

### 2.7. Analysis of Amino Acid and Free Amino Acids Content

The analysis of the amino acid of the sample (laver and surimi) was carried out according to the amino acid analysis method of the Korean Food Standards Codex [27]. A 0.20 g of the ground sample was accurately weighed and placed in a test tube. Next, 15 mL of 6N HCl was added, and the sample was hydrolyzed at 110 °C in a dry oven for 24 h. The hydrolyzed sample was filtered with a glass filter and then concentrated under vacuum at 55 °C using a rotary evaporator. Sample was then diluted to 25 mL with pH 2.20 citric buffer and filtered with a 0.45 µm membrane filter (Hyundai Micro, Seoul, Republic of Korea) to prepare the sample for amino acid analysis. A 2 g mass of freeze-dried sample was mixed with 30 mL of 70% ethanol and then centrifuged. The precipitate was then mixed with another 30 mL of 70% ethanol, and the supernatant was concentrated under vacuum (<40 °C). The sample extracted and concentrated with ether was diluted with citric buffer, and then 1 g of sulfosalicylic acid was added and mixed. The mixture was then allowed to stand in a dark room for 1 h, and then filtered with a 0.45 μm membrane filter to prepare the sample for the free amino acid analysis.

The composition and free amino acids analysis were performed using an amino acid analyzer (Sykam amino acid analyzer S433, Sykam GmbH, Eresing, Germany). The column and the standard solution were LCA K07/Li 4.6 × 150 mm and standard solution amino acids (Sykam GmbH, Eresing, Germany), respectively.

### 2.8. Texture Profile

The texture of the laver chip was measured using a texture analyzer (TA-XT2 Plus, Stable micro systems Ltd., Godalming, UK) following the method of Stamataki et al. [28]. The laver chips used for this measurement were 25 mm (w) × 40 mm (l) × 10–15 mm (h) in size. The brittleness was measured using a three-point bend rig (small size) probe. The texture analyzer condition was set and conducted as follows: Pre-test speed: 5.00 mm/s; test speed: 5.00 mm/s; post-test speed: 5.00 mm/s; force: 100 N; trigger force: 0.049 N; probe height: 35 mm; probe distance: 15 mm (*n* ≧ 3). 

### 2.9. Puffing Ratio Analysis

The puffing ratio of the laver chip was measured by the seed replacement method using a waxy millet [29]. The puffing ratio was calculated using the following Equation (1).
Puffing ratio (%) = (V_2_ − V_1_/M_0_) × 100(1)

M_0_: Dry weight (g), V_1_: Volume before puffing (mL), V_2_: Volume after puffing (mL).

### 2.10. Flavor Profiles Analysis

The analysis of the volatile aromatic components of the sample was conducted according to the method of Cha et al. [30]. The adsorption of flavor compounds was carried out using a solid-phase micro extraction (SPME) device (Supelco, Inc., Bellefonte, PA, USA). The adsorption fiber was a polydimethylsiloxane/divinylbenzene (PDMS/DVB) fiber (65 μm coating thickness). It was activated in the GC injection port at 220 °C for 30 min prior to analysis. A 6 mL sample and 1 μL (101.14 ng) of the internal standard hexyl acetate (Sigma Co., St. Louis, MO, USA) were added to a 20 mL headspace glass vial (Supelco, Inc., Bellefonte, PA, USA). The vial was sealed with an aluminum crimp seal (20 mm, open center) and polytetrafluoroethylene (PTFE)/silicone septum (60 mils). The fiber was exposed to the vial for 60 min at 50 °C to extract volatile flavor compounds. The fiber was exposed to the GC injection port at 220 °C for 5 min to desorb the volatile compounds. The extraction of volatile components by the SPME method was performed three times for each sample.

The volatile flavor compounds adsorbed by the SPME method were desorbed for 5 min in the injection port of a PerkinElmer 600T GC/MSD (PerkinElmer Co., Fremont, CA, USA). The analysis column was an Elite Wax capillary column (60 m × 0.25 mm i.d. × 0.25 μm film thickness, Perkin Elmer Co., Fremont, CA, USA). The carrier gas (helium) was run at a linear velocity of 1.0 cm/s. The oven temperature was held at 40 °C for 5 min, then increased up to 220 °C at a rate of 4 °C/min, and held for 10 min. The MSD analysis condition was as follows: capillary direct interface temperature, 220 °C; ion source temperature, 204 °C; ionization energy, 70 eV; mass range, 33 to 350 a.m.u; electron multiplier voltage, 1500 V [31]. The tentative identification of each compound was performed by searching the retention index (RI) and the NIST (The National Institute of Standards and Technology, version 2.3) MS library database (Perkin Elmer Co., Fremont, CA, USA). The quantification of the identified volatile flavor compounds was calculated as the relative content (factor = 1, ng/g) using the internal standard (hexyl acetate).

### 2.11. Sensory Analysis

The sensory evaluation of the laver chip was approved by the Institutional Review Board of Changwon National University (IRB No. 7001066-202211-HR-071). The sensory evaluation (preference test) of RFIS for the laver chip was conducted on 48 participants including undergraduate students, graduate students, and faculty members of the Department of Food Science and Nutrition at Changwon National University using the ranking test method (1 point for the most preferred; 4 points for the least preferred). In addition, the overall acceptance of the samples (the laver chip and RFIS) was evaluated by 13 panelists using a 9-point scale, and 10 of them were then used to conduct a quantitatively descriptive analysis of the samples. The sensory panelists were given training on the definition, principles, and methods of QDA, as well as the concepts of the types and intensities of sensory attributes. The training was conducted for about 2 months, three times a week. Next, each sensory evaluation panelist was asked to suggest their sensory descriptive language for the samples. After discussion, if more than half of the panelists agreed, the sensory descriptive language was determined. Finally, 6 odor profiles were selected for RFIS, including baked potato, savory, sweet, nutty, meaty, and fish sauce. The standard substances for each odor profile were selected as follows: baked potato-like odor was 3-methylthiopropanal (Sigma-Aldrich Co., St. Louis, MI, USA), sweet was ethylmaltol (Anhui Jinhe Ind. Co. Ltd., Chuzhou, China), and savory, nutty, meaty, and fish sauce were butter-roasted squid (Keumhan, Gangneung, Gangwon, Republic of Korea), roasted almond (Dongwoonongsan, Gwangju, Republic of Korea), bacon (Chungjungone Co., Seoul, Republic of Korea), and fish sauce product (Daeyoung Fishery Food, Changwon, Republic of Korea), respectively, which were purchased from a local market. The strength of each odor was evaluated by comparing it to the standard substance using a 9-point scale (1 point: very weak, 9 points: very strong).

### 2.12. Statistical Analysis

In this study, the experiment was repeated (*n* ≧ 3), and the results were presented as mean ± standard deviation. In addition, the results were analyzed using ANOVA (analysis of variance) with SPSS software (IBM SPSS Statistics 27, IBM, New York, NY, USA). The significance of the difference between the mean values of each measurement was verified using Duncan’s multiple range test at the *p* < 0.05 level.

## 3. Results and Discussion

### 3.1. Amino Acid Contents of Major Ingredients (Laver and Surimi)

The results of the amino acid composition of the laver and surimi are shown in Table 2. The total amino acid content of laver was 31,610 mg/100 g, of which the essential and non-essential amino acid contents were 11,293 mg/100 g (35.73%) and 20,316 mg/100 g (64.27%), respectively, suggesting that it is an excellent source of protein. In addition, six essential amino acids, including lysine, threonine, leucine, isoleucine, valine, and phenylalanine, were detected. Among them, leucine, valine, and isoleucine, which are widely known as BCAA (branched-chain amino acids), accounted for a high proportion with contents of 2454.94 mg/100 g, 2360.26 mg/100 g, and 1462.11 mg/100 g, respectively. This suggests that the laver is an excellent source of BCAA. In addition, nine non-essential amino acids, including arginine, alanine, aspartic acid, glutamic acid, glycine, histidine, serine, proline, and tyrosine, were detected. Among them, glutamic and aspartic acids had the highest levels with contents of 3624.97 mg/100 g and 3289.47 mg/100 g, respectively. On the other hand, the total amino acid content of dried laver from different regions of Republic of Korea was reported to range from 156 to 310.9 mg/g [32].

In the case of surimi, the total amino acid content was 18,398.07 mg/100 g, of which the essential and non-essential amino acid contents were 7089.31 mg/100 g and 11,308.76 mg/100 g, respectively. Similar to the result of laver, six essential amino acids, including lysine, threonine, leucine, isoleucine, valine, and phenylalanine, were detected. Among them, the contents of lysine and BCAA (leucine, isoleucine, and valine) were very high. In addition, eight non-essential amino acids, including arginine, alanine, aspartic acid, glutamic acid, glycine, serine, proline, and tyrosine, were detected. Content of glutamic acid was the highest (3199.26 mg/100 g) exhibiting the same trend as that of the laver. Based on a dry basis, the amino acid content of surimi (75,155.51 mg/100 g) was more than twice that of dried laver (34,018.53 mg/100 g). Therefore, it was thought that surimi, when used as a major ingredient of the laver chip along with dried laver, would be an excellent source of amino acids.

Next, the total free amino acid content of laver was 3171.91 mg/100 g, which was 10% of the total amino acids. According to a previous study [33] that classified the characteristics of amino acids according to taste, the contents of aspartic and glutamic acids, which are known as umami amino acids, were 1235.95 mg/100 g, the contents of 5 amino acids (threonine, serine, glycine, alanine, lysine) known as sweet amino acids were 421.4 mg/100 g, and the contents of 6 amino acids known as bitter amino acids, including valine, methionine, isoleucine, leucine, phenylalanine, and histidine, were 41.41 mg/100 g. The ratios of these to the total free amino acid content were sequentially 38.97%, 13.29%, and 1.31%, respectively, with the umami amino acid accounting for the largest portion. In addition, L-glutamic acid, which is one of the umami amino acids, accounted for 35.00% (1110.22 mg/100 g), which was second highest after taurine (41.82%), and it was found to be the main factor in the flavor component of dried laver. However, the content of threonine, methionine, and glycine as precursor substances for generating savory and palatable odors (baked potato-like, nutty, and meaty) was relatively low.

### 3.2. Preliminary Experiment for Determining the Optimum Composition Ratio of Laver Chip Batter

A total of four preliminary experiments were conducted to determine the optimum composition ratio of the major (laver, surimi, and rice flour) and minor ingredients (leavening agent, soybean oil, and corn syrup) for processing the laver chip batter. The results of these experiments are shown in Figure 2. Firstly, the addition ratios of the main ingredients, laver and surimi, were compared and analyzed at a mass ratio (%, *w*/*w*) of 10:10 (%) to 40:40 (%). The remaining part of the batter was added at a ratio of rice flour (%, *w*/*w*) (Figure 2A). The optimum composition ratio of the main ingredients showed a significant increase in brittleness as the laver and surimi content increased, or as the rice flour content decreased (*p* < 0.05). However, there was no significant difference in the puffing ratio between all samples, but the ratio of 20:20 g% (Spl 2) was the highest. This trend was consistent with that presented in the study by [34], which found that the puffing ratio decreased with decreasing the added amount of rice flour to fish snacks made with Pollock. In addition, three types of snacks (Ref 1–3) with high preference in the market were measured under the same experimental condition, and the brittleness values ranged from 6.31 to 12.09 N. However, panel members preferred a soft texture with a strength of 6 N. Therefore, the condition with the highest puffing ratio in the range of 6–8 N brittleness, i.e., the selected ratio of laver: surimi: rice flour was 20:20:60%, and the next stage of the experiment was conducted.

In the following experiment on the additional amount of leavening agent (NaHCO_3_), there was a concern that the product would have a bitter taste as the concentration of NaHCO_3_ increased [35]. Therefore, GDL, which is tasteless and odorless [35], was fixed at 3 g, and the results of analyzing the texture and puffing ratio by increasing the NaHCO_3_ content from 0 to 2 g were shown in Figure 2B. The puffing ratio increased with increasing concentration of NaHCO_3_ up to 1.5 g (*p* > 0.05). In addition, the brittleness also showed a decreasing trend with increasing additional amount. Therefore, the ratio of GDL: NaHCO_3_ = 3 g:1.5 g was selected, which has the highest puffing ratio and the lowest brittleness.

The following was the result of measuring the changes in puffing ratio and brittleness of the laver chip according to the additional amount of soybean oil (2~6 g%) to reduce the physical strength (brittleness) and process laver chip softer. The results were shown in Figure 2C. The brittleness of the laver chip significantly decreased with increasing the additional amount of soybean oil (*p* < 0.05). However, there was no significant difference in brittleness (5.16~5.79 N) when the additional amount of soybean oil was 4 g or more. In the case of the puffing ratio, there was no significant difference between all samples, ranging from 111.83 to 113.97%. In conclusion, the addition of 4 g of soybean oil resulted in low brittleness and a high puffing ratio. Jeong and Lee [36] found that the brittleness of Chamchwi (*Aster scaber*) snacks decreased with an increasing amount of soybean oil, but the oiliness also increased, leading to a negative sensory impact.

Next, the changes in brittleness and puffing ratio of the laver chip according to the additional amount of corn syrup (4–12 g%) for flavoring the laver chip were shown in Figure 2D. Corn syrup is known to have a lower processing cost and a sweeter taste than those of refined sugar [37]. Therefore, the appropriate addition range of the corn syrup was determined based on sensory evaluation. The brittleness of the laver chip increased with increasing amounts of corn syrup, while the puffing ratio did not show a significant trend within the range of 111.68 to 113.89 N (*p* < 0.05). In the case of adding 8 g of corn syrup, the brittleness was less than 7 N. In the sensory domain, the sweetness was evaluated as high when the additional amount of corn syrup was 8 g or more. Therefore, the additional amount of corn syrup in this experiment was selected as 8 g.

To summarize the above results, the batter composition for processing the laver chip pellets was composed of 20% of laver, 20% of surimi, and 60% of rice flour. In addition, 3 g of GDL, 1.5 g of NaHCO_3_, 4 g of soybean oil, and 8 g of corn syrup were added as leavening agents and sweetener. In addition to the major components, 1 g of salt and 0.01 g of saccharin were added, and 140 mL of distilled water was used to process the batter for the laver chip.

### 3.3. Determination of the Optimum Composition of Laver Chip Batter through RSM

The optimum composition of protein-rich laver chip batter through RSM was based on the results of preliminary experiments. Specifically, as shown in Table 1 (RSM 1), the ratio of laver and surimi was set and coded in 5 stages, from 20:20 to 28% (*w*/*w*) with increasing proportions of surimi, which has relatively high protein content. Air-frying time was set to 40–60 s in an air fryer (195 °C). The experimental design was then developed using the central composite design (CCD) method. In particular, 14 experiments were randomly conducted with 4 fractional points (No. 1–4), 4-star points (No. 5–8), and 6 central points (No. 9–14). The dependent variables were the brittleness and puffing ratio values.

In terms of brittleness, the lowest value (6.16 N) was obtained for No. 6, which had 1.6 times the amount of surimi added to laver (20:26%) and was air-fried for 60 s. No. 8 (20:24%, 40 s of air-frying) showed the highest value (10.12 N). In terms of appearance and sensory evaluation, the laver chip air-fried for 40 s was not completely cooked resulting in a tough texture. Accordingly, the brittleness was also relatively high (No. 8). On the other hand, the laver chip air-fried for 60 s was burnt and had a bitter taste (No. 6) resulting in the lowest brittleness.

In terms of puffing ratio, No. 5 (20:28% = laver: surimi, 50 s of air-frying) had the highest value (116.01%), while No. 2 (20:26% = laver: surimi, 45 s of air-frying) had the lowest value (114.33%). In all samples (No. 1–14), the puffing ratio ranged from 114.33 to 116.01%, with a variation of less than 1.46%. There was also no significant difference in appearance. The results of the multiple regression analysis of the dependent variables (brittleness and puffing ratio) obtained according to the central composite design are shown in Table 3.

In the case of brittleness, when the laver content was fixed at 20% (*w*/*w*) and the surimi addition amount was increased from 20 to 28%, both the linear term (X_1_) and air-frying time (X_2_) were significant (*p* < 0.05) in the first-order equation, but only X_2_ was significant in the second-order equation, and all cross products were not significant. On the other hand, in the case of the puffing ratio, the two independent variables (X_1_, X_2_) did not show significance in any of the first-order, second-order, or cross-product terms. In the case of brittleness, the lack of fit test and R2 values were 0.132 and 0.9897, respectively, which were satisfactory (*p* > 0.05). However, in the case of puffing ratio, the values were 0.014 and 0.2728, respectively. Therefore, the design model based on the dependent variable, the puffing ratio was not suitable.

Therefore, the optimization was performed with the goal of minimizing brittleness (<7 N) and achieving the average value of the central point measurement value (115.02% of puffing ratio). As a result, the optimization batter composition (laver: surimi: rice flour) and processing condition (air-frying time) were 20:21.3:58.7% (*w*/*w*) and 52.50 s, while the predicted values of brittleness and puffing ratio were 6.5973 N and 115.03%, respectively.

### 3.4. Optimization of the Composition Ratio of RFIS though RSM

The central composite design was developed from the coded design in Table 1 (RSM 2). A total of 17 experiments were randomly performed, consisting of 8 fractional points, 6 star points, and 3 central points. The dependent variables were the measurements of baked potato-like and savory odors, which were the most dominant odors among the six odors identified by QDA. In the case of the baked potato-like odor, the sensory test results obtained from 13 QDA panelists using a 9-point scale showed a range from 3.78 to 6.22 throughout the entire experimental range. Among them, the highest value of 6.22 was observed in No. 16 (methionine 0.10 g (*w*/*v*), threonine 0.70 g (*w*/*v*), glycine 1.30 g (*w*/*v*)) and the lowest value of 3.78 was obtained in No. 10 (methionine 0.06 g (*w*/*v*), threonine 0.90 g (*w*/*v*), glycine 0.55 g (*w*/*v*)). On the other hand, in the case of savory odor, it showed a range from 1.89 to 4.40 throughout the entire experimental range. Among them, the highest value of 4.40 was observed in No. 1 (methionine 0.14 g (*w*/*v*), threonine 0.50 g (*w*/*v*), glycine 0.55 g (*w*/*v*)) and No. 2 (methionine 0.14 g (*w*/*v*), threonine 0.90 g (*w*/*v*), glycine 1.05 g (*w*/*v*)) and the lowest value of 1.89 was obtained in No. 9 (methionine 0.10 g (*w*/*v*), threonine 0.70 g (*w*/*v*), glycine 0.30 g (*w*/*v*)). The results of the multiple regression analysis of the dependent variables (baked potato-like and savory odors) obtained according to the central composite design are shown in Table 4. It was not significant in the case of savory odor either. The results of the response surface model equations for baked potato-like and savory odors (ANOVA) showed that the coefficient of determination (R^2^) and lack of fit values were 0.7989 and 0.648 for baked potato-like odor and 0.4614 and 0.818 for savory odor. This indicates that the model equation for baked potato-like odor is more appropriate (Table 4). Based on the RSM results, the optimum composition ratio that satisfies the maximum value of baked potato-like (Y_1_) and savory (Y_2_) odors was methionine (X_1_) 0.18 g, threonine (X_2_) 1.10 g, and glycine (X_3_) 0.80 g, respectively. Based on the optimum composition ratio, the predicted values of baked potato-like and savory odors were 6.5321 and 4.6537, respectively.

In addition, sniffing test results showed that the reaction odor had baked potato-like, savory, and sweet odors confirming the effectiveness of the application of the reaction flavor. Ko et al. [19] also reported that a model experiment in which methionine and glucose were reacted at 100 °C emitted boiled potato and sweet odors.

### 3.5. GC-MS Analysis and Identification Results of the Processed RFIS

The results of the analysis of the volatile flavor components of the RFIS produced with the optimum composition ratio obtained from RSM and the total amount of compounds by group are shown in Table 5 and Figure 3. A total of 41 volatile aroma components were detected and identified with the following group distribution: 8 aldehydes, 6 ketones, 6 S-containing compounds, 6 N-containing compounds, 6 furans, 3 alcohols, 2 acids, 2 esters, and 2 miscellaneous compounds.

In the aldehyde group, 3-methylthiopropanal, which has a baked potato-like odor, was the most abundant compound detected. This compound had a low threshold of 0.2 ppb [38], suggesting that it was the dominant odor component of RFIS in this experiment. Next, 5-methyl-2-furancarboxaldehyde (threshold: 6 ppb, Izzreen et al. [39], which has a baked or caramelized odor, was detected in a large amount. Decanal (green-like, fruity) and nonanal (green-like, fruity, Cha et al. [40]) followed.

In the S-containing compounds group, the content of dimethyl disulfide (threshold: 12 ppb, Cha et al. [40]), which has a cooked cabbage or crab meat-like odor, was dominant. The next highest contents were dimethyl trisulfide (threshold: 0.01 ppb, Cha et al. [40]) and methanethiol (threshold: 2.21 ppb, Zhu et al. [41]) having a cooked cabbage odor. These compounds were thought to be the dominant odor components of RFIS because they have very low thresholds. On the other hand, in the furans group, furfural (threshold: 282 ppb, Wang et al. [42]), which has sweet and almond-like odors, was the most abundant compound detected. The next highest contents were 2-methyl-5-[(methylthio) methyl] furan and 2,4-dimethylfuran. At the N-containing compound group, only trimethylpyrazine (threshold: 23 ppb, Counet et al. [43]), which has cocoa, roasted, and green odors, was detected at a concentration of 10.20 ng/g. Based on the comparison of the threshold and detection levels of each compound, S-containing compounds (dimethyl disulfide, dimethyl trisulfide, methanethiol) and aldehydes (3-methylthiopropanal, 5-methyl-2-furancarboxaldehyde, decanal, nonanal) were considered to be the main odor components of RFIS induced by reaction flavor precursors (amino acids and glucose).

### 3.6. Determination of RFIS Concentration and Additional Amount for Processing Laver Chip

To determine the appropriate RFIS concentration and an additional amount for processing laver chip, the ranking test was applied with increasing concentrations of amino acids (methionine, threonine, glycine, glutamic acid) and glucose obtained through RSM by one time (RFIS 1), two times (RFIS 2), three times (RFIS 3), and five times (RFIS 5). The results of the sensory test (ranking test) are shown in Table 6. RFIS 3 was the best, followed by RFIS 5, RFIS 2, and RFIS 1. However, RFIS 2–5 showed no significant difference, while RFIS 1 showed a significant difference (*p <* 0.05). In the case of RFIS 3, the baked potato-like and sweet odors were well-balanced. However, in the case of RFIS 5, the sweet odor was too strong (melted sugar-like odor), and the baked potato-like odor was relatively masked. This was consistent with the increasing content of furan compounds in Figure 3.

On the other hand, the concentrations of aldehyde and furan groups increased relatively with increasing additional amount (*p* < 0.05), while the S-containing compound group showed the highest content at RFIS 3 and then decreased at RFIS 5. In addition, the ketone, alcohol, acid, ester, and N-containing compound groups had very low contents so it was thought that their contribution to the flavor of RFIS was very low. Therefore, it was thought that the compounds generated from RFIS in this study, such as 3-methylthiopropanal, dimethyl trisulfide, and dimethyl disulfide, will act as aroma-active compounds or positive components in the laver chip. In addition, it was thought that these compounds will have the effect of masking the off-flavor of the laver itself.

### 3.7. Optimum Processing Condition of Laver Chip Obtained through RSM

The predicted and actual values of brittleness and puffing ratio of the laver chip, processed with the optimum processing condition obtained through RSM, were 6.5973 N and 115.03% and 6.93 ± 0.33 N and 116.19 ± 0.48%, respectively. These were within a 10% error rate for both brittleness and puffing ratio. In addition, the predicted and actual values of baked potato-like and savory odors of RFIS obtained through RSM were 6.5321 and 4.6537 and 6.00 ± 0.78 and 4.00 ± 0.91, respectively, according to QDA testing (9-point scale). It was confirmed that the baked potato-like and savory odors of RFIS were both within a 10% error rate.

Therefore, the optimized processing condition obtained through RSM were as follows: The major ingredients, laver (20%), surimi (21.3%), and rice flour (58.7%), were mixed together. GDL (3 g), NaHCO_3_ (1.5 g), soybean oil (4 g), corn syrup (8 g), salt (1 g), saccharin (0.01 g), 126 mL of distilled water, and 14 mL of RFIS were added to the mixture as minor ingredients. The shaped laver chip pellet was dried at 50 °C (1–2 h) and then air-fried in the air-fryer (195 °C and 52.5 s). At this time, the RFIS was processed by reacting methionine (0.54% *w*/*v*), threonine (3.30% *w*/*v*), glycine (2.40% *w*/*v*), glutamic acid (0.90% *w*/*v*), and glucose (3% *w*/*v*) in 100 mL of distilled water at 121 °C for 90 min.

The moisture and crude protein, fat, and ash contents of the laver chip processed with the optimum composition and the processing condition were 5.51 ± 0.02 g% and 24.26 ± 0.10 g%, 2.19 ± 0.05 g%, and 4.30 ± 0.07 g%, respectively. While commercial laver chips typically contain 3 to 6% protein, the high-protein laver chip developed in this study had an enhanced protein content of 24%.

Therefore, it was possible to process the laver chip with a very high-protein content, gluten-free, and low calorie (371.56 kcal) by applying air-frying and reaction flavor technologies.

## 4. Conclusions

The objective of this study was to determine the optimum processing condition for high-protein, gluten-free, and low-calorie laver chip by applying RSM to process the laver chip via air-frying and reaction flavor technologies. The optimum composition ratio of the batter through RSM was 20% (*w*/*w*) laver, 21.3% (*w*/*w*) surimi, and 58.7% (*w*/*w*) rice flour. To achieve this, GDL 3 g, NaHCO_3_ 1.5 g, soybean oil 4 g, corn syrup 8 g, salt 1 g, saccharin 0.01 g, 126 mL of distilled water, and 14 mL of RFIS were added and mixed as the minor ingredients to process a batter. The batter was then dried (50 °C and 1–2 h) to produce pellets. The pellets were then air-fried in an air-fryer (195 °C and 52.5 s). The measured values of brittleness and puffing ratio of the laver chip were 6.93 ± 0.33 N and 116.19 ± 0.48%, respectively, which were all within 10% of the predicted values. In addition, the RFIS was produced by adding the precursor materials (methionine 0.54% (*w*/*v*), threonine 3.30% (*w*/*v*), glycine 2.40% (*w*/*v*), glutamic acid 0.90% (*w*/*v*), and glucose 3% (*w*/*v*)) to the distilled water (100 mL) and heated (121 °C and 90 min). The measured QDA (baked-potato and savory odors) values were 6.00 ± 0.78 and 4.00 ± 0.91, respectively. These were also within 10% of the predicted values. The major flavor components of the laver chip were identified as aldehydes (3-methylthiopropanal, 5-methyl-2-furancarboxaldehyde, decanal, nonanal) and S-containing compounds (dimethyl disulfide, dimethyl trisulfide, methanethiol). The laver chip produced in this study had a high-protein content (24.26 g%) and was low in calories (371.56 kcal). In addition, further research will be needed on the physical properties of air-fried chips according to the heating conditions of air-frying in the processing of air-fried chips.

## Figures and Tables

**Figure 1 foods-12-04450-f001:**
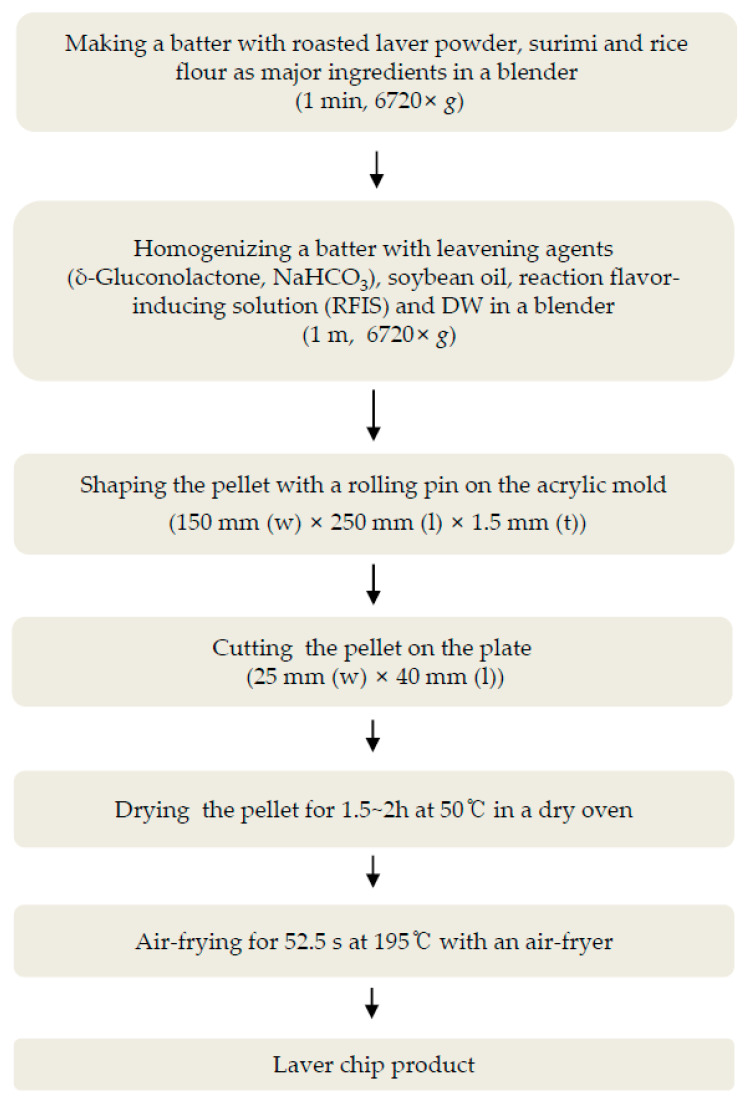
Flow chart for the preparation of laver chip.

**Figure 2 foods-12-04450-f002:**
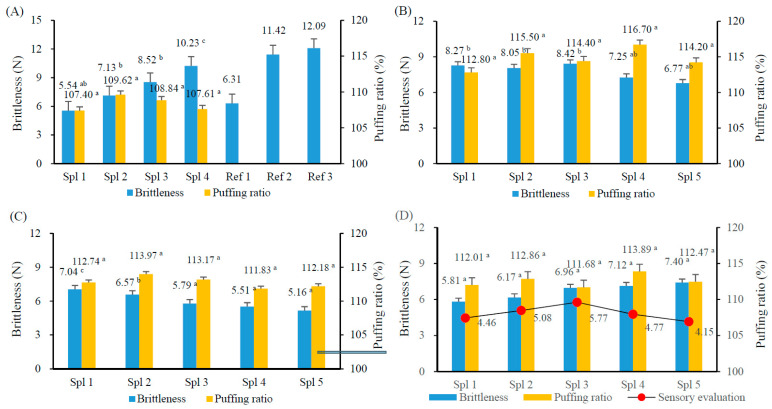
Brittleness, puffing ratio and sensory evaluation (**D**) of laver chip according to the amount of ingredients. (**A**): Spl (sample) 1, laver and surimi 10:10 g%; Spl 2, 20:20 g%; Spl 3, 30:30 g%; Spl 4, 40:40 g% (*w*/*w*) Rice flour ratio(%) was set 100-(laver and surimi ratio), and ref (reference, commercial snacks). (**B**): Spl 1, GDL 3 g only, and Spl 2, 3, 4, and 5 were 3 g of GDL plus with NaHCO3 0.5 g, 1.0 g, 1.5 g, and 2.0 g under condition of the ratio of laver (20%): surimi (20%): rice flour (60%), respectively. (**C**): Spl 1, soybean oil (SO) 2 g, and Spl 2, 3, 4, and 5 were 3, 4, 5, and 6 g of SO under condition of the ratio of laver (20%): surimi (20%): rice flour (60%), and GDL 3 g+ NaHCO3 1.5 g, respectively. (**D**): Spl 1, corn syrup (CS) 4 g, and Spl 2, 3, 4, and 5 were 6, 8, 10, and 12 g of CS under condition of the ratio of laver (20%): surimi (20%): rice flour (60%), and GDL 3 g%+ NaHCO3 1.5 g%, soybean 4 g%, respectively. Drying and frying conditions for making laver chip: Drying at 50 °C for 1–2 h and frying at 195 °C for 50 s. ^a–c^ Means with the same superscript are not significantly different by Duncan’s test (*p* < 0.05).

**Figure 3 foods-12-04450-f003:**
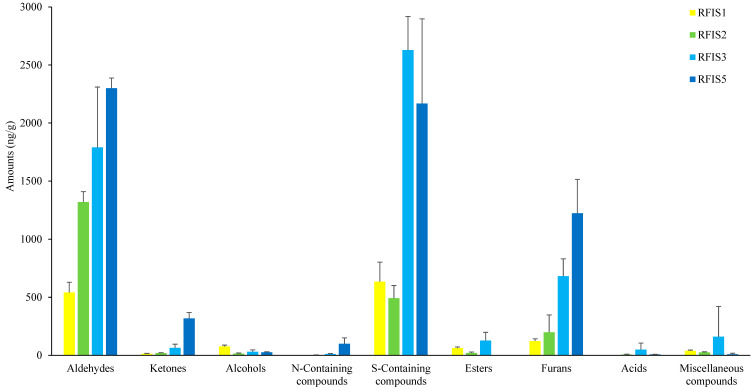
The amounts of classified volatile compounds detected in reaction flavor solution induced with precursor compounds.

**Table 1 foods-12-04450-t001:** Coded and uncoded independent variables used in RSM design for making laver chip.

		Symbol	Independent Variable				
			−2	−1	0	+1	+2
RSM 1 ^1^	Laver and surimi ratio (g%)	X_1_	20:20	20:22	20:24	20:26	20:28
	Frying time (s)	X_2_	40	45	50	55	60
RSM 2 ^2^	Methionine (*w*/*v*, %)	X_1_	0.02	0.06	0.10	0.14	0.18
	Threonine (*w*/*v*, %)	X_2_	0.30	0.50	0.70	0.90	1.10
	Glycine (*w*/*v*, %)	X_3_	0.30	0.55	0.80	1.05	1.30

^1^ Rice flour ratio (%) was set to 100-(laver + surimi ratio); ^2^ Glutamic acid and glucose were set with 0.3% and 1.0% to 100 mL of distilled water in advance.

**Table 2 foods-12-04450-t002:** Amino acid compositions of dried laver and surimi and free amino acid composition of dried laver as major components, (mg/100 g).

Amino Acid (A.A)	Dried-Laver	Surimi
Lysine **	1823.70 ± 6.07 (1961.90) ^1^	1861.14 ± 8.08 (7602.69) ^1^
Threonine **	1842.22 ± 27.13 (1982.59)	905.88 ± 7.21 (3700.49)
Leucine *^,^**	2454.94 ± 5.84 (2641.99)	1538.11 ± 12.03 (6283.13)
Isoleucine *^,^**	1462.11 ± 11.00 (1573.51)	1003.62 ± 2.83 (4099.75)
Valine *^,^**	2360.26 ± 17.18 (2540.10)	1011.97 ± 13.47 (4133.86)
Phenylalanine **	1350.15 ± 30.62 (1453.02)	768.59 ± 25.30 (3139.67)
Tyrosine ***	1074.80 ± 24.45 (1156.69)	608.08 ± 7.54 (2483.99)
Arginine ***	2322.33 ± 24.20 (2499.28)	1359.74 ± 15.32 (5554.49)
Histidine ***	1025.99 ± 12.81 (1104.65)	676.79 ± 5.27 (2764.67)
Glutamic acid ***	3624.97 ± 53.47 (3901.17)	3199.26 ± 28.13 (13,068.87)
Glycine ***	2154.22 ± 28.42 (2318.36)	732.31 ± 4.91 (2991.46)
Serine ***	1533.69 ± 34.12 (1650.55)	715.12 ± 4.82 (2921.24)
Proline ***	1767.27 ± 30.70 (1901.93)	723.67 ± 33.25 (2956.17)
Alanine ***	3533.89 ± 44.27 (3803.15)	1078.17 ± 7.84 (4404.29)
Aspartic acid ***	3289.47 ± 48.54 (3540.10)	2215.62 ± 19.12 (9050.74)
Essential A.A	11,293.38 (12,153.47)	7089.31 (28,959.60)
Non-essential A.A	20,316.64 (21,864.66)	11,308.76 (46,195.92)
Total	31,610.02 ± 251.33 (34,018.53)	18,398.07 ± 131.29 (75,155.51)
**Free amino acid (F.A.A)**	**Dried-laver**	
L-Lysine ^4^	16.50 ± 1.61 (17.76)	
L-Threonine ^4^	8.61 ± 0.54 (9.27)	
L-Leucine ^5^	11.53 ± 0.62 (12.41)	
L-Isoleucine ^5,^*	8.52 ± 0.48 (9.17)	
L-Valine ^5,^*	14.44 ± 0.88 (15.54)	
L-Methionine ^5^	tr. ^2^	
L-Phenylalanine ^5^	6.92 ± 0.76 (7.48)	
L-Tryptophan	tr.	
L-Cystine	tr.	
L-Tyrosine	6.28 ± 0.15 (6.76)	
L-Arginine	tr.	
L-Histidine ^5^	tr.	
L-Glutamic acid ^3^	1110.22 ± 2.97 (1194.81)	
L-Aspartic acid ^3^	125.73 ± 4.65 (135.31)	
L-α-Aminoadipic acid	tr.	
L-α-Aminobutyric acid	tr.	
β- Aminobutyric acid	tr.	
γ-Aminobutyric acid	12.82 ± 0.57 (13.80)	
L-Carnosine	tr.	
L-Citrulline	23.36 ± 2.62 (25.14)	
1-Methyl-L-histidine	tr.	
3-Methyl-L-histidine	tr.	
L-Ornithine	tr.	
Asparagine	4.55 ± 0.62 (4.90)	
Hydroxyproline	tr.	
Phosphoethanolamine	tr.	
L-Glycine ^4^	6.43 ± 0.36 (6.92)	
L-Serine ^4^	6.02 ± 0.52 (6.48)	
L-Alanine ^4^	383.84 ± 4.14 (413.09)	
L-Proline	tr.	
Phosphoserine	99.65 ± 4.06 (107.24)	
Taurine	1326.49 ± 24.20 (1427.56)	
Urea	tr.	
β-alanine	tr.	
Total bitter amino acids	41.41 (44.57)	
Total sweet amino acids	421.4 (453.51)	
Total umami amino acids	1235.95 (1330.12)	
Total	3171.91 ± 28.15 (3413.59)	

* Branched Chain A.A, ** Essential A.A, *** Non-essential A.A; ^1^ Dry basis in parentheses. ^2^ Not detected; ^3^ Umami: aspartic acid + glutamic acid; ^4^ Sweet: threonine + serine + glycine + alanine + lysine; ^5^ Bitter: valine + methionine + isoleucine + leucine + phenylalanine + histidine.

**Table 3 foods-12-04450-t003:** Model coefficients estimated by multiple linear regression of dependent variables (brittleness and puffing ratio) for processing laver chip.

Factors	Coefficients	
	Brittleness (N)	Puffing Ratio (%)
Constant	7.5409	114.949
	Linear	
[Laver and surimi ratio]	0.3350 *	0.100
[Frying time]	−0.9833 *	0.093
	Quadratic	
[Laver and surimi ratio]^2^	−0.0278	0.072
[Frying time]^2^	0.1435 *	0.091
	Cross product	
[Laver and surimi ratio] × [Frying time]	−0.0300	−0.025
Model		
Linear	0.000	0.648
Quadratic	0.003	0.640
Cross product	0.685	0.922
R-square	0.9897	0.2728
Total model	0.000	0.833
Lack of fit	0.132	0.014

** p* < 0.05.

**Table 4 foods-12-04450-t004:** Model coefficients estimated by multiple linear regression of dependent variables (baked potato and savory odor) for processing RFIS.

Factors	Coefficients	
	Baked Potato Odor	Savory Odor
Constant	5.395	3.304
	Linear	
[Met]	0.337 *	0.332
[Thr]	0.102	0.033
[Gly]	0.424 *	0.189
	Quadratic	
[Met]^2^	−0.087	−0.056
[Thr]^2^	−0.082	−0.028
[Gly]^2^	5.395	−0.126
	Cross product	
[Met] × [Thr]	0.234	0.239
[Met] × [Gly]	−0.141	−0.379
[Thr] × [Gly]	0.209	0.109
Model		
Linear	0.648	0.014
Quadratic	0.640	0.080
Cross product	0.922	0.307
R-square	0.799	0.461
Total model	0.721	0.075
Lack of fit	0.648	0.819

** p* < 0.05.

**Table 5 foods-12-04450-t005:** Volatile flavor compounds detected in reaction flavor-inducing solution (RFIS) (ng/g).

Aldehydes (8) (540.95 ± 88.29) ^3^ (1789.73 ± 520.43) ^4^	Ketones (6) (13.45 ± 3.08) ^3^ (63.72 ± 31.73) ^4^
3-Methylthiopropanal (918.00 ± 273.18) ^1^, 5-Methyl-2-furancarboxaldehyde (421.18 ± 101.43) ^2,^ Decanal (338.87 ± 73.84), Benzaldehyde (0.00 ± 0.00), 3-Methylbenzaldehyde (0.00 ± 0.00) ^2^, Octanal (0.00 ± 0.00), Nonanal (86.66 ± 63.90), 1-Methly-1H-pyrrole-2-carboxaldehyde * (25.02 ± 8.08) ^2^	2,3-Butanedione (15.68 ± 2.56) ^2^, 1-(Ethylthio)-2-propanone * (16.40 ± 14.24) ^2^, 2-Hydroxy-3-methyl-2-cyclopenten-1-one * (0.00 ± 0.00) ^2^, 6,10-Dimethyl-5,9-undecadien-2-one * (0.00 ± 0.00), 4-(2-furanyl)-3-buten-2-one * (0.00 ± 0.00) ^2^, 1-(1H-pyrrol-2-yl)-ethanone * (31.64 ± 14.93) ^2^
**S-Containing compounds (6) (634.84 ± 168.04) ^3^ (2628.53 ± 289.74) ^4^**	**N-Containing compounds (6) (0.00 ± 0.00) ^3^ (10.20 ± 5.52) ^4^**
Methanethiol (50.76 ± 9.90) Dimethyl disulfide (2488.27 ± 250.69) Dimethyl trisulfide (68.26 ± 22.03) (Methylthio)cyclopentane (21.24 ± 7.12) Trimethylthio disulfide (0.00 ± 0.00) (Methylthio)cyclohexane (0.00 ± 0.00) ^2^	2,5-Dimethylpyrazine (0.00 ± 0.00) ^2^, 2,6-Dimethylpyrazine (0.00 ± 0.00) ^2^, 2-Ethyl-6-methylpyrazine (0.00 ± 0.00) ^2^, Trimethylpyrazine (10.20 ± 5.52) ^2^, 3,4-Dihydro-6-methyl-2H-pyran * (0.00 ± 0.00) ^2^, 3-Ethyl-2,5-dimethylpyrazine (0.00 ± 0.00) ^2^
**Furans (6) (121.95 ± 20.38) ^3^ (660.73 ± 142.41) ^4^**	**Alcohol (3) (76.60 ± 10.83) ^3^ (30.41 ± 16.72) ^4^** **and Esters (2) (62.02 ± 10.21) ^3^ (126.38 ± 71.58) ^4^**
Dihydro-2-methyl-3(2H)-furanone (0.00 ± 0.00) ^2^, Furfural (552.20 ± 119.61), 2-[(Methylthio)methyl]furan (0.00 ± 0.00), 2-Methyl-5-[(methylthio)]methyl]furan * (108.53 ± 22.80) ^2^, 2,4-Dimethylfuran * (21.27 ± 7.23) ^2^, 5-Hydroxymethylfurfural * (0.00 ± 0.00) ^2^	2-Furanmethanol (0.00 ± 0.00), Dodecanol (30.41 ± 16.72), Phenol (0.00 ± 0.00) ^2^, Bis(2-methylpropyl) hexandioate * (26.14 ± 11.37), 2-Ethylhexyl benzoate * (100.24 ± 60.21)
**Acids (2) (0.00 ± 0.00) ^3^ (49.13 ± 56.51 ^4^**	**Miscellaneous compounds (2) (38.33 ± 7.11) ^3^ (159.55 ± 261.73) ^4^**
Acetic acid (0.00 ± 0.00) ^2^, Nonanoic acid (49.13 ± 56.51) ^2^,	D-Limonene (0.00 ± 0.00), Naphthalene (159.55 ± 261.73)

* Compounds were tentatively identified by mass spectrum (NIST data) only. ^1^ Amount in parenthesis was detected in RFIS 3 which was added 3 times the concentration of RFIS 1 and then the other processes were the same as that of RFIS 1 (*n* = 3). ^2^ Compound was detected below 0.01 ng in RFIS 1 which was added amino acids (glutamic acid 0.3 g, methionine 0.18 g, threonine 1.10 g, and glycine 0.8 g) and glucose 1.0 g into D.W 100 mL, and then reacted at 121 °C for 90 min. ^3^ Total amounts in parentheses were detected in RFIS 1 (*n* = 3). ^4^ Total amounts in parentheses were detected in RFIS 3 (*n* = 3).

**Table 6 foods-12-04450-t006:** Result of sensory evaluation for optimum adding concentration of reaction flavor-inducing solution (RFIS) for processing laver chip.

	RFIS 1 ^2^	RFIS 2 ^2^	RFIS 3 ^2^	RFIS 5 ^2^
Overall acceptance	141 ^a^	116 ^b^	108 ^b^	115 ^b^
Ranking ^1^	4th	3rd	1st	2nd

^1^ Ranking test (*n* = 48): (1st = 1, 2nd = 2, 3rd = 3, and 4th = 4 score); ^2^ RFIS 1: Added amino acids (glutamic acid 0.3 g, methionine 0.18 g, threonine 1.10 g, and glycine 0.8 g) and glucose 1.0 g into D.W 100 mL, and then reacted at 121 °C for 90 min. RFIS 2, 3, and 5: Added 2, 3, and 5 times of the concentration of RFIS 1, respectively, and then the other processes were the same as that of RFIS 1. ^a,b^ Means with the same superscript are not significantly different by Duncan’s test (*p* < 0.05).

## Data Availability

The data presented in this study are available in article.

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
