# Peer review of "Determination of Optimum Processing Condition of High Protein Laver Chip Using Air-Frying and Reaction Flavor Technologies"

_foods, 2023, doi:10.3390/foods12244450_

Round 1

Reviewer 1 Report

Comments and Suggestions for Authors

The manuscript aimed to process laver chip using air-frying and reaction flavor technologies and to identify the optimum processing condition for a new type of laver chip that was high in protein, gluten-free, and low in calories, which were different from conventional oil fried laver products. The manuscript contains some several scientific concerns:

1.Lines 365-367, how are significant differences identified? Is the significance in Figure 2A Brittleness or Puffing ration? Moreover, no significant difference can be seen in Fig. 2A. Why not make a significant comparison with the control in Fig. 2A?

2. In Fig. 1A, there is a significant difference between the Brittleness of Ref1 and Ref2 and 3? Have you ever noticed which kind of sales in the market is relatively high (Ref1, 2 or 3)? Is it consistent with the result of your group's preference, "panel members preferred a soft texture with a strength of 6 N" in Line 373?

3. The deviation of partial data in Fig. 3 is too large. Is the result credible?

4. Is there a list of essential nutrients for the Laver Chip? Because the title is about "high protein High Laver Chip", there should be a nutrition facts list that compares to the same type of product on the market (especially content of crude protein), so that it can highlight the high protein of your product.

5. What are the advantages of your product compared to similar products on the market today? What are the innovations of the manuscript?

Comments on the Quality of English Language

The manuscript aimed to process laver chip using air-frying and reaction flavor technologies and to identify the optimum processing condition for a new type of laver chip that was high in protein, gluten-free, and low in calories, which were different from conventional oil fried laver products. The manuscript contains some several scientific concerns:

1.Lines 365-367, how are significant differences identified? Is the significance in Figure 2A Brittleness or Puffing ration? Moreover, no significant difference can be seen in Fig. 2A. Why not make a significant comparison with the control in Fig. 2A?

2. In Fig. 1A, there is a significant difference between the Brittleness of Ref1 and Ref2 and 3? Have you ever noticed which kind of sales in the market is relatively high (Ref1, 2 or 3)? Is it consistent with the result of your group's preference, "panel members preferred a soft texture with a strength of 6 N" in Line 373?

3. The deviation of partial data in Fig. 3 is too large. Is the result credible?

4. Is there a list of essential nutrients for the Laver Chip? Because the title is about "high protein High Laver Chip", there should be a nutrition facts list that compares to the same type of product on the market (especially content of crude protein), so that it can highlight the high protein of your product.

5. What are the advantages of your product compared to similar products on the market today? What are the innovations of the manuscript?

Author Response

Please see the attached document for our response to your comments.

Reviewer 2 Report

Comments and Suggestions for Authors

Referee report for
"Determination of Optimum Processing Condition of High Protein Laver Chip using Air-Frying and Reaction Flavor Technologies"
By: Gyeong-Tae Jeong, Changheon Lee, Eunsong Cha, Seungmin Moon, Yong-Jun Cha, and Daeung Yu
This study looked into the use of reaction flavoring and air-frying technologies for the processing of laver chip, South Korean seafood.   Response Surface Methodology by Central Composite Design was used to find the optimal ratio of the three primary ingredients (laver, surimi, and rice flour) for a novel kind of laver chip.   The paper is novel in that it prepares a reaction flavor-inducing solution and laver chips utilizing the air-frying method.   The manuscript's main body is coherent, and the paper is well-organized.   The outcomes are attractive, and the required materials and methods were also thoroughly explained.   The manuscript's English language proficiency is good;   however, there are a few errors.   Further, there are some hesitations in the methodology.   Need a minor revision,my suggestions are included below for the authors to take into account.
English language errors:
1.      Line 17 (page 1): "with air-fried" should be "with air-frying".
2.      Many sentences are started with "And".   Please remove it.   After a "full stop", another sentence cannot be initiated with "and".   For exemple: Line 131 (page 3), 139 (page 3), 171 (page 5), 211 (page 6), 327 (page 8), 488 (page 13), and 527 (page 14).
3.      Line 129-130 (page 3). The sentences are repetitive.   Please remove them.
4.      Line 182 (page 5): "Among" should be "among".
5.      Line 216 (page 6): "the sample" should be "sample".
6.      Line 310 (page 6): please add "respectively" before "suggesting".
7.      Line 359 (page 10): "Preliminary" should be "preliminary".
Technical comments and questions
8.      It is better to unify the units such as "hour, minute, and second" throughout the paper.   In some cases, the authors used "sec., min., hr" and in others "hour, minute, and second".

9.      Line 315 (page 8): the authors mentioned that: 9 non-essential amino acids were detected in laver chip.   But, in the next sentence only the name of 8 amino acids were listed as follows: arginine, alanine, aspartic acid, glutamic acid, glycine, serine, proline, and tyrosine.

10.      Line 400 (page 10): The reference "Kim et al., 2015" must be in numerical format "[37]".

11.      In the "Figure 1.   Flow chart for the preparation of laver chip", the authors mentioned that: first a batter is made with the combination of roasted laver powder, surimi, and rice flour in a blender.   But, in the lines 127-133 (page 3), it was mentioned that: "First, batter was made by mixing roasted laver powder and surimi in a 1:1 ratio and blending for 1 minute with the blender.   Then, leavening agents (GDL and NaHCO3), soybean oil, rice flour, and RFIS were 131 added in distilled water in a certain amount according to the condition".   These methods are not the same.   Please, specify the method as you used in your experiments.

12.      For preparation of the reaction flavor-inducing solution (RFIS) (lines 121-122), you used the data from preliminary test and optimization.   Do you have any reference to mention?

13.      In lines 131-132, "the leavening agents (GDL and NaHCO3), soybean oil, rice flour, and RFIS were added in distilled water in a certain amount according to the condition".   Please specify the weight of each element and the volume of the distilled water.   Because, it affects the moisture content of the laver chip before oven drying and the other results.

14.      In Figure 1, why the volume of laver batter before and after adding the leaving agents are the same (6720 xg)?

15.      Please mention the initial moisture content of the laver chip batter before oven drying?   Is it determined or not?

16.      Lines 182-189, you used RSM to determine the optimum mixing ratio of RFIS for processing laver chip.   But, glutamic acid and glucose were not considered in your optimization process and they were optimized by other researchers in other papers.   Your research condition was different than the work of Cha and Wang [25] and Ahn et al. [21].   Why you did not do the optimization with the presence of glutamic acid and glucose?

Comments on the Quality of English Language

Author Response

(The authors gave the same response as above.)

Round 2

Reviewer 1 Report

Comments and Suggestions for Authors

Through the modification, the article has been improved greatly. This article is recommended for acceptance.